# Cardiac Biomarkers in a Brazilian Indigenous Population Exposed to Arboviruses: A Cross-Sectional Study

**DOI:** 10.3390/v16121902

**Published:** 2024-12-10

**Authors:** Jandir Mendonça Nicacio, Carlos Dornels Freire de Souza, Orlando Vieira Gomes, Beatriz Vasconcelos Souza, João Augusto Costa Lima, Rodrigo Feliciano do Carmo, Sávio Luiz Pereira Nunes, Vanessa Cardoso Pereira, Naiara de Souza Barros, Ana Luiza Santos de Melo, Lucca Gabriel Feitosa Lourencini, Jurandy Júnior Ferraz de Magalhães, Diego Guerra de Albuquerque Cabral, Ricardo Khouri, Manoel Barral-Netto, Anderson da Costa Armstrong

**Affiliations:** 1Faculty of Medicine, Federal University of Vale do São Francisco—UNIVASF, Petrolina 56304-917, PE, Brazil; carlos.dornels@univasf.edu.br (C.D.F.d.S.); orlando.gomes@univasf.edu.br (O.V.G.); rodrigo.carmo@univasf.edu.br (R.F.d.C.); 2Postgraduate Program in Human Ecology and Socio-Environmental Management, Bahia State University—UNEB, Juazeiro 48904-711, BA, Brazil; cardosovanessap@gmail.com; 3Postgraduation Program in Epidemiology and Health Problems Control, Oswaldo Cruz Foundation/Fiocruz, Recife 50670-420, PE, Brazil; 4Postgraduate Program in Human Pathology, Faculty of Medicine of Bahia, Federal University of Bahia, Salvador 40026-010, BA, Brazil; beatriz.vasconcelos@fiocruz.br; 5Cardiology, Johns Hopkins University, Baltimore, MD 21205, USA; jlima@jhmi.edu; 6Postgraduate Program in Applied Cellular and Molecular Biology, University of Pernambuco-UPE, Recife 50100-010, PE, Brazil; savionunes12@gmail.com; 7Collegiate of Medicine, Faculty of Medicine, Federal University of Vale do São Francisco—UNIVASF, Petrolina Campus, Petrolina 56304-917, PE, Brazil; naiara.souzab@discente.univasf.edu.br (N.d.S.B.); ana.melo@discente.univasf.edu.br (A.L.S.d.M.); lucca.lourencini@discente.univasf.edu.br (L.G.F.L.); 8College of Medicine-Serra Talhada Campus-UPE/ST, University of Pernambuco, Serra Talhada 56909-205, PE, Brazil; jurandy.magalhaes@upe.br; 9Agamenon Magalhães Hospital, Serra Talhada 50751-530, PE, Brazil; 10Central Public Health Laboratory of Pernambuco—LACEN, Recife 50050-215, PE, Brazil; diego.g.a.cabral@hotmail.com; 11Oswaldo Cruz Foundation/Fiocruz, Institute Gonçalo Moniz, Salvador 40296-710, BA, Brazil; ricardo_khouri@hotmail.com (R.K.); manoel.barral@fiocruz.br (M.B.-N.); 12School of Medicine, Federal University of Bahia—UFBA, Salvador 40170-110, BA, Brazil; 13Rega Institute for Medical Research, KU Leuven, 3000 Leuven, Belgium; 14Instituto Nacional de Ciência e Tecnologia de Investigação em Imunologia, University of São Paulo, São Paulo 05347-902, SP, Brazil

**Keywords:** arboviruses, cardiac involvement, dengue, chikungunya, Zika, cardiac biomarkers

## Abstract

Arthropod-borne viral diseases are acute febrile illnesses, sometimes with chronic effects, that can be debilitating and even fatal worldwide, affecting particularly vulnerable populations. Indigenous communities face not only the burden of these acute febrile illnesses, but also the cardiovascular complications that are worsened by urbanization. A cross-sectional study was conducted in an Indigenous population in the Northeast Region of Brazil to explore the association between arboviral infections (dengue, chikungunya, and Zika) and cardiac biomarkers, including cardiotrophin 1, growth differentiation factor 15, lactate dehydrogenase B, fatty-acid-binding protein 3, myoglobin, N-terminal pro-B-type natriuretic peptide, cardiac troponin I, big endothelin 1, and creatine kinase-MB, along with clinical and anthropometric factors. The study included 174 individuals from the Fulni-ô community, with a median age of 47 years (interquartile range 39.0 to 56.0). High rates of previous exposure to dengue, chikungunya, and Zika were observed (92.5%, 78.2%, and 95.4% anti-IgG, respectively), while acute exposure (anti-IgM) remained low. The biomarkers were linked to age (especially in the elderly), obesity, chronic kidney disease, and previous or recent exposure to chikungunya. This study pioneers the use of Luminex xMAP technology to reveal the association between cardiac inflammatory biomarkers and exposure to classical arboviruses in an Indigenous population undergoing urbanization.

## 1. Introduction

Arthropod-borne viral diseases are emerging, debilitating acute febrile illnesses with serious long-term consequences that threaten nearly 4 billion people in tropical and subtropical areas [1]. West Nile virus, Zika virus, dengue, and chikungunya virus are examples of emerging viruses that can cause chronic diseases, mainly when they affect the central nervous or musculoskeletal system, such as neurological syndromes, chronic joint pain, arthritis, and congenital Zika syndrome [1,2,3,4,5]. These diseases impact various cultural and socioeconomic groups across different regions, particularly individuals with comorbidities such as hypertension, diabetes, heart disease, and obesity [6,7]. Historically more prevalent in tropical areas, their incidence has expanded in recent decades to temperate climates, affecting both urban populations and rural or traditional communities [8]. Once neglected and underestimated, these diseases now exhibit both endemic and epidemic patterns, causing acute systemic manifestations, often with severe and debilitating cardiac involvement. In this context, dengue, chikungunya, and Zika pose the greatest challenges to public health [2,9,10].

Dengue fever is a significant global health issue that affects all continents and puts about half of the world’s population at risk. It results in an estimated 400 million infections and 22,000 deaths annually [11]. In August 2024, Brazil reported over 6.5 million probable cases, more than twice as many as in 2023, with an incidence rate of 3205 per 100,000 inhabitants [12]. Furthermore, chikungunya has been reported in over 100 countries, with Brazil recording 254,562 probable cases and an incidence rate of 125.4 per 100,000 inhabitants in the same period [12,13]. Chikungunya virus disease has been associated with an increased risk of death for up to 84 days after symptom onset, including deaths from cerebrovascular diseases, ischemic heart diseases, and diabetes [14]. The Zika virus, responsible for Zika fever, emerged in 2015 with major outbreaks in South America associated with microcephaly. It was declared an international health emergency by the World Health Organization in 2016. By August 2024, Brazil had reported over 6568 probable cases, with an incidence rate of 3.2 per 100,000 inhabitants [12,15].

Both urban and rural traditional communities, especially Indigenous populations, are among the groups most affected by arboviral diseases and the urbanization process [16]. In Brazil, approximately 1,694,836 self-identified Indigenous people belong to 305 ethnic groups and speak 274 languages. The majority (58.9%) live in rural areas, while 41.1% reside in urban regions, and they represent nearly 1% of Brazil’s total population [17]. Even those who live predominantly in rural areas are affected by urbanization, as proximity to urban centers affects their way of life and significantly impacts the health of their communities [18], leading to an increase in cardiovascular diseases caused by acute febrile illnesses or chronic non-communicable diseases [19,20,21,22].

Cardiovascular diseases are chronic non-communicable diseases that significantly impact public health, and they are the leading cause of mortality worldwide, with over 17 million deaths per year. This is especially true for people over 70 years of age. Their impact is also greater in low-income populations and vulnerable communities, which account for more than three-quarters of these deaths [23,24]. In the context of arboviruses, cardiovascular disease is a concerning issue, as the number of infected Indigenous people is increasing, leading to a higher risk of cardiovascular complications [25,26].

The role of arboviruses as potential triggers or contributors to cardiovascular disease, including subclinical conditions, has been investigated in recent years, but their impact on vulnerable Indigenous communities remains poorly understood. However, arboviruses are known to be associated with both direct and indirect damage to the myocardium, along with increased expression of cytokines such as interleukin 6 (IL-6), interleukin 13 (IL-13), interleukin 18 (IL-18), interleukin 1 beta (IL-1β), and tumor necrosis factor alpha (TNF-α) [21,22]. In addition, biomarkers such as growth differentiation factor 15 (GDF-15), N-terminal pro-B-type natriuretic peptide (NT-proBNP), and high-sensitivity C-reactive protein (hs-CRP) have shown considerable importance in the detection and management of cardiovascular diseases, especially heart failure, in patients affected by various diseases [27,28,29].

Despite the growing number of studies on cardiovascular disease prediction [30], studies focusing on serum biomarkers in populations with limited access to health services have yet to be completed. They show solid predictive potential for risk, especially when correlated with obesity, kidney disease, gender, age, and lifestyle [31]. However, more research is needed in this area, particularly in Indigenous populations. These groups are at different stages of demographic and epidemiological change, and there is still a limited understanding of how they develop cardiovascular disease, especially given the different influences of the urbanization process.

The biomarkers investigated include inflammatory cytokines, peptides, enzymes, and proteins. CT-1, for example, is a member of the IL-6 cytokine family that was originally characterized as a factor that induces cardiomyocyte hypertrophy and survival (cytoprotective and regulatory effects) and specifically protects against ischemic damage [28,32]. GDF-15, in turn, is a cytokine of the transforming growth factor beta family that is present in small amounts in tissue and plasma. Its expression is upregulated by stress and tissue damage and is associated with inflammatory conditions of the myocardium. It is also a cardioprotective protein and a biomarker for cardiovascular risk [33].

Lactate dehydrogenase B (LDH-B), a subunit of the lactate dehydrogenase enzyme, plays a central role in cardiac energy metabolism by utilizing lactic acid, a by-product of anaerobic glycolysis, as an energy substrate during myocardial ischemia. This process underscores its cardioprotective role under ischemic conditions, possibly attenuating cardiac hypertrophy [34,35,36]. Fatty-acid-binding protein 3 (FABP-3), predominantly expressed in cardiomyocytes, facilitates intracellular lipid transport and metabolism. FABP-3 is released into the bloodstream when the sarcolemma is damaged, such as in ischemia, and is an early biomarker of cardiac injury [37]. Myoglobin, a monomeric heme protein in cardiac and skeletal muscle, is a highly sensitive marker of myocardial damage, particularly in acute ischemia. In addition to its role in oxygen transport, it regulates nitric oxide and reactive oxygen species, emphasizing its pathophysiological importance [38].

NT-proBNP (N-terminal pro-brain natriuretic peptide), an inactive fragment of the prohormone released by ventricular cardiomyocytes under wall stress, is widely recognized as a guideline-recommended biomarker of heart failure. Elevated NT-proBNP levels indicate increased cardiac stress and are associated with poor prognosis in patients with heart failure [39,40]. These biomarkers provide a multifaceted view of cardiac health and information about metabolic, structural, and functional cardiac changes in different pathophysiological states.

Cardiac troponin I (cTnI), encoded by the TNNI3 gene, is a highly specific marker of myocardial injury, including acute myocardial infarction. Elevated cTnI levels indicate cardiomyocyte damage, as emphasized in the American College of Cardiology and American Heart Association guidelines [41]. Similarly, big endothelin-1 (big ET-1), the biologically inactive precursor of endothelin-1, exerts significant effects by reducing cardiac output, promoting myocardial hypertrophy, and stimulating collagen synthesis in cardiac fibroblasts [42].

This study aims to describe the cardiac biomarkers in an Indigenous population exposed to arboviruses (dengue, chikungunya, and Zika). Exposure was assessed through serological markers (IgM and IgG), indicating past or recent infections. The study population comprised individuals from a traditional Indigenous community, distinguished by their cultural and historical ties to the region, and did not include recent migrants or newcomers to the area. Additionally, the study explores the relationships between these biomarkers and clinical, demographic, and anthropometric characteristics to better understand the unique profile of this population.

## 2. Materials and Methods

### 2.1. Study Design and Population

This is a descriptive, cross-sectional study conducted as an ancillary study of the first phase of the Project of Atherosclerosis among Indigenous Populations (PAI), which occurred between August 2016 and June 2017. The PAI study protocol has been previously described [43]. Briefly, the PAI study was initiated in 2016 as an observational study whose primary objective was to assess cardiovascular health in two Indigenous groups with different levels of urbanization (Fulni-ô and Truka Peoples) and a fully urbanized control group in the city of Juazeiro, Bahia, Brazil [43].

For the current analysis, we only included individuals from the Fulni-ô community. The Brazilian Institute of Geography and Statistics (IBGE, acronym in Portuguese) has differentiated this population on the basis of anthropological, ethnolinguistic, socio-demographic, geographical, cultural, and self-identifying characteristics [44]. The Fulni-ô People live in the municipality of Águas Belas along the banks of the Ipanema River, a tributary of the São Francisco River, in the Agreste region of the state of Pernambuco, Brazil. This area covers 12,000 square meters and is home to 5627 people who show signs of limited urbanization. They preserve their Indigenous language, which is taught in schools, and maintain cultural and religious practices, including at least three months of seclusion a year, during which non-Indigenous people are prohibited from entering [43] (Figure 1).

This study included all individuals over 29 years of age who received care through the PAI. The research and healthcare visits took place between August 2016 and June 2017, during which clinical, demographic, and anthropometric data were collected by completing a structured form (Appendix A).

The following exclusion criteria were applied: clinical manifestations of heart failure, history of acute coronary events requiring hospitalization, dialysis-dependent renal failure, history of surgery for peripheral artery disease or heart disease, cerebrovascular diseases requiring hospitalization, and participants with restrictions on blood collection and storage.

### 2.2. Anthropometric Parameters and Clinical Data

Data on age, sex, weight, and body mass index (BMI) were collected from the Fulni-ô Indigenous community following the PAI research protocol [43]. BMI classification adhered to World Health Organization standards as follows: underweight (<18.5), normal (18.5 to 24.9), overweight (25 to 29.9), and obese (≥30) [45]. Participants self-reported hypertension diagnoses and medication use, with details available in a Appendix A. Blood pressure was measured three times using an Omron^®^ BP785 IntelliSense^®^ monitor (Kyoto, Japan), following Brazilian Society of Cardiology guidelines. Hypertension was defined as systolic blood pressure ≥ 140 mmHg, diastolic blood pressure ≥ 90 mmHg, or use of hypertension medication [46]. Diabetes was identified by an HbA1c level of 6.5% or higher or diabetes medication use [47]. The estimated glomerular filtration rate was calculated using the CKD-EPI creatinine equation, excluding the variable of race [48]. Participants were classified into the following six categories: G1—normal/high (≥90 mL/min/1.73 m^2^), G2—mildly decreased (60 to 89 mL/min/1.73 m^2^), G3a—mildly to moderately decreased (45 to 59 mL/min/1.73 m^2^), G3b—moderately to severely decreased (30 to 44 mL/min/1.73 m^2^), G4—severely decreased (15 to 29 mL/min/1.73 m^2^), and G5—kidney failure (<15 mL/min/1.73 m^2^) [47].

The research process respected Indigenous religious practices and rights. Individuals with serious health problems, including those who did not meet the study’s inclusion criteria, were cared for by a specialized medical team and referred to treatment centers to promote community health. Thus, the working team conducted research and focused on providing comprehensive care to the community.

### 2.3. Study Variables

The following variables were analyzed: sociodemographic factors (sex and age); obesity (BMI ≥ 30 kg/m^2^); systemic arterial hypertension; diabetes mellitus; chronic kidney disease; serology for arboviruses (dengue, chikungunya, and Zika); and the following cardiac markers: cardiotrophin 1 (CT-1), growth differentiation factor 15 (GDF-15), lactate dehydrogenase B (LDH-B), fatty-acid-binding protein 3 (FABP-3), myoglobin, NT-proBNP, cardiac troponin I (cTnI), big endothelin 1 (ET-1), and creatine kinase MB (CKMB).

### 2.4. Serum Collection for Biomarker Analysis and Arbovirus Serology

Cardiac biomarkers were assessed using the Cardiac Disease 9-Plex Human ProcartaPlex^TM^ kit (EPX090-15809-901, Waltham, MA, USA), which enables cardiovascular research by analyzing the following nine protein biomarkers: CT-1, GDF-15, LDH-B, FABP-3, myoglobin, NT-proBNP, cTnI, big ET-1, and CKMB. This was achieved through Luminex xMAP technology (multi-analyte profiling), which utilizes differentially dyed capture beads for each target in a multiplex ELISA-like assay. The ProcartaPlex immunoassays allow the simultaneous analysis of multiple proteins in a single sample, providing an efficient and accurate method for detecting cytokines and cardiac biomarkers [49,50].

Peripheral blood sampling was performed by venipuncture in the right or left antecubital fossa. After appropriate antisepsis, 5 to 10 mL of venous blood was collected and transported in refrigerated boxes at 2 to 8 °C. The serum was then centrifuged in 1.5 mL Eppendorf tubes and stored at −20 to −70 °C. These samples were stored at the Instituto Gonçalo Moniz, FIOCRUZ Bahia, and at the Clinical Analysis Laboratory, LPC, Salvador, Bahia.

Serological tests for anti-dengue virus IgG/IgM; anti-chikungunya virus IgG/IgM; and anti-Zika virus IgG/IgM were performed using enzyme-linked immunosorbent assay (ELISA) kits (Euroimmun, code: EI 293a-9601G) according to the manufacturer’s instructions. The assay utilized tetramethylbenzidine as the substrate for colorimetric detection, and optical density (OD) was measured using a spectrophotometer at 450 nm [51]. The relative index was calculated as the ratio of the sample OD to the cutoff OD provided by the kit. Values with a relative index ≥ 1.1 were considered positive; values ≥ 0.8 and <1.1 were considered borderline; and values < 0.8 were considered negative [51]. To investigate Zika seroprevalence (anti-IgG ZIKV), the manufacturer used the non-structural protein NS1 to reduce cross-reactions. A cut-off of 20 RU/mL is recommended (Euroimmun, 2017). Regarding the accuracy of the test used, the Euroimmun anti-CHIKV ELISA IgG assay (EUROIMMUN AG, Lübeck, Germany) showed 95.4% sensitivity and 100% specificity in validation studies on another specific population, in previous studies [51].

### 2.5. Statistical Analysis

A descriptive analysis of the study population was performed to characterize continuous and categorical variables. The Shapiro–Wilk test was employed to assess the normality of the distribution for continuous variables. The non-parametric Mann–Whitney U test was applied for group comparisons upon identifying a non-Gaussian distribution. Continuous variables were summarized using measures of central tendency (median) and dispersion (interquartile range), while categorical variables were presented as absolute values and proportions. Logistic regression models using generalized least squares were applied to evaluate the association between serological status for arbovirus (IgM and IgG) and inflammatory and cardiac biomarkers, when appropriate. The regression models estimated the effect size (coefficients) for each biomarker, adjusting for potential confounders. Estimates were reported with standard errors, z-values, and *p*-values. The variance inflation factor was utilized to assess multicollinearity. Residual analysis was conducted to verify model assumptions, including normality, linearity, and homoscedasticity. Residuals showed no significant deviations from normality, confirming model adequacy. All statistical analyses were conducted using JASP software (Jeffreys’s Amazing Statistics Program), version 0.16.1. A significance level of 5% (*p* < 0.05) was adopted for all statistical inferences.

### 2.6. Ethical Approval

The study received approval from the National Council on Research Ethics (CONEP, acronym in Portuguese) under number 1.488.268, the National Indigenous Peoples’ Foundation (FUNAI, acronym in Portuguese) under procedure number 08620.028965/2015-66, and the Indigenous leaders of the participating groups. All participants provided written informed consent before participating in the study.

## 3. Results

The study included a total of 174 individuals from the Fulni-ô community, with a median age of 47 (interquartile range 39.0 to 56.0). The majority of the participants fell within the age range of 30 to 49 years (28.2%), and 18.4% (n = 32) were aged 60 years or older. The study found that 29.3% (n = 51) of the participants were obese according to the World Health Organization standards, with a slightly higher prevalence among women (31.6% compared to 25.0% for men), but this difference was not statistically significant (*p* = 0.365). Additionally, 17.8% (n = 31) had systemic arterial hypertension, with a higher prevalence in women compared to men (22.8% and 8.3%, respectively; *p* = 0.018). In the total population studied, 10.3% (n = 18) had diabetes mellitus. Nearly 30% (n = 51) were obese, and 3.4% (n = 6) had chronic kidney disease, with no significant difference between the sexes (Table 1).

### 3.1. Seroprevalence of Arboviruses

A high rate of exposure to the serologic profile of classical arboviruses was observed in this community. The prevalence of anti-DENV IgG was 92.5% (161/174), anti-CHIKV IgG was 78.2% (136/174), and anti-ZIKV IgG was 95.4% (166/174). With regard to the detection of acute exposure to arboviruses (identified by the presence of IgM antibodies), a low prevalence was found, distributed in increasing order as follows: anti-DENV IgM, 4.6%; anti-CHIKV IgM, 1—2.6%; and anti-ZIKV IgM, 16.1%, with no significant differences between sexes (Table 1). It is important to highlight that the study was conducted in a municipality situated in a hot and arid region. Although the study period coincided with the meteorological winter, there was a dry season followed by increased precipitation and rainfall starting in March 2017. Additionally, monitoring of Aedes aegypti larval infestation levels in households, carried out using the periodic sampling technique known as the Rapid Survey Infestation Index (LIRAa), reported a LIRAa index of 2.0 in the city of Águas Belas, a value classified as alert status [52,53].

### 3.2. Cardiac Biomarkers

Of the nine biomarkers analyzed, two were associated with sex (higher myoglobin levels in men and higher NT-proBNP levels in women), two with age (higher GDF-15 and big ET-1 levels in older adults), two with obesity (higher NT-proBNP and CKMB levels in participants who were obese), and one with chronic kidney disease (big ET-1). Conversely, four biomarkers showed higher serum levels in those without diabetes (CT-1, LDH-B, FABP-3, and myoglobin). In addition, no markers were associated with systemic arterial hypertension. Concerning arboviruses, no markers were associated with exposure to dengue and Zika, whether acute or past (IgG). Two markers, CT-1 and LDH-B, were associated with the presence of IgM for chikungunya, and two markers, LDH-B and big ET-1, were associated with the presence of IgG for chikungunya. (Table 2). Furthermore, LDH-B was not associated with comorbidities or age, but it showed significant associations with both the acute and chronic phases of chikungunya virus infection and participants without diabetes, according to the nonparametric analysis.

Figure 2 displays the cardiac and inflammatory biomarkers associated with explanatory variables, including arbovirus exposure. Additionally, detailed measures of central tendency and dispersion for these markers are provided in the Appendix A. The data show significant variations in the levels of markers such as CT-1, LDH-B, FABP-3, myoglobin, and big ET-1 between the groups studied, highlighting the complexity and variability of biological responses in different health conditions (Appendix A).

A logistic regression model was used to analyze the biomarkers that showed statistical significance in serological CHIKV status. The results showed a consistent association between CT-1 and acute exposure to the virus (IgM), while LDH-B was associated with previous exposure (IgG), as shown in Table 3.

## 4. Discussion

This report highlights the significant impact of dengue, chikungunya, and Zika on the Fulni-ô population, a partially urbanized Indigenous group in the Northeast Region of Brazil, with seroprevalences exceeding 90% for dengue and Zika, and nearly 80% for chikungunya. These rates are remarkably high compared to most studies in the Americas, where prevalences are generally below 50% [54,55].

The high seroprevalence of arboviruses, particularly flaviviruses, in the Fulni-ô Indigenous community reflects the endemicity of these infections in the region, which maintain detectable IgG levels due to their long-term persistence. While IgM antibodies were detected in a lower proportion of individuals (5–16%), this pattern is consistent with a population that is predominantly in the convalescent or post-infection phase [56]. Additionally, cultural and behavioral practices, such as limited mobility outside their territory and religious activities in groups, may increase exposure to *Aedes aegypti* vectors, thereby contributing to the cumulative seroprevalence [32]. To better understand this epidemiological pattern, future studies should include longitudinal monitoring of antibody dynamics, georeferenced mapping of transmission hotspots, identification of circulating arbovirus serotypes, and enhanced vector surveillance focusing on seasonal variations. These approaches could provide valuable insights into the interplay between endemicity and community-specific risk factors.

For instance, a 2015–2016 study in a socially vulnerable community in the Northeast Region Brazil found a chikungunya prevalence of 11.8% [54], while another study in an urbanized population reported 12.8% for Zika and 50.8% for dengue [55]. Comparable high seroprevalences have only been observed in a few studies, such as 75% for chikungunya in Comoros in 2004 [57] and 74.6% for dengue in Brazil [58], typically in vulnerable and urbanizing communities. On the other hand, seroprevalence studies in Indigenous and traditional communities are much rarer. In a recent study in the Indigenous Orang Asli community in Malaysia, anti-DENV, anti-CHIKV, and anti-ZIKV IgG antibodies were detected in 46.1%, 47.3%, and 29.7% of individuals, respectively [59].

In contrast to our findings, official data from neighboring non-Indigenous municipalities (Águas Belas, Itaíba, and Iati) report a low incidence of suspected arbovirus cases based on clinical assessments between 2014 and 2016 [60]. This stark epidemiological contrast remains unexplained. Potential factors, for example, differences in healthcare access, diagnostic coverage, underreporting, environmental exposure, and social behaviors, may contribute to the observed disparities. However, the exact dynamics require further investigation.

### 4.1. Cardiac Biomarkers, Demographic Variables, and Comorbidities

In our study, significant sex-specific differences in cardiac biomarkers were observed in this Indigenous population exposed to arboviruses. Myoglobin levels were primarily associated with males, possibly due to their higher engagement in physical labor, consistent with findings by Dannecker et al. (2012) that indicate sex-related variations in muscle damage markers [61]. In contrast, NT-proBNP levels were higher in females, likely linked to the greater prevalence of hypertension and obesity in Indigenous women, conditions known to cause cardiac wall stress and volume overload [62,63]. Hormonal factors may also contribute to these differences; testosterone may have a protective cardiovascular effect in men, whereas estrogen and conditions such as polycystic ovary syndrome could elevate NT-proBNP in women [64]. These findings underline the importance of sex-specific factors and hormonal influences in cardiovascular risk assessment for Indigenous populations [65,66].

Another important finding of this study was the significant association between age and the biomarkers GDF-15 and big ET-1 in older individuals. GDF-15, a key regulator in aging, is known for its roles in reducing systemic inflammation, supporting mitochondrial function, and managing energy homeostasis, thus acting as a biomarker for age-related conditions and cardiovascular and renal diseases [67,68,69,70]. Similarly, elevated levels of big ET-1 in older adults and individuals with chronic kidney disease highlight its importance as a biomarker for cardiovascular diseases, including atherosclerosis and hypertension, as well as its pathogenic role in chronic kidney disease due to its potent vasoconstrictive properties [71,72,73].

Furthermore, our study showed elevated NT-proBNP and CKMB levels in obese individuals and provides new insights into cardiac biomarkers in this population. In contrast to the existing literature, which often reports lower NT-proBNP levels in obese patients due to increased clearance or altered secretion, our results show the opposite [74,75]. However, other concomitant factors that produce increased cardiac stress or subclinical dysfunction and affect the results of this study must also be considered. Similarly, CKMB, a recognized marker of myocardial necrosis, may be elevated in obese individuals, reflecting increased cardiac workload [76]. These findings emphasize the need for careful interpretation of NT-proBNP and CKMB in clinical practice and possible adjustment of diagnostic thresholds for obesity to improve the diagnosis and treatment of heart failure.

### 4.2. Cardiac Biomarkers, Arboviruses, and Comorbidities

Arboviruses are acute diseases with chronic effects. The severity of their clinical outcome may be associated with comorbidities such as hypertension, diabetes, chronic kidney disease, and advanced age [77,78]. However, as the design of this study did not allow us to identify arboviruses by clinical severity, we were unable to investigate this association.

When exploring the relationships between biomarkers and the serological profile for arboviruses, we found that CT-1 and LDH-B were associated with the acute phase of chikungunya (anti-CHIKV IgM). CT-1, a member of the IL-6 cytokine family, is known for its role in cardiac hypertrophy and inflammation [79]. Although specific studies on CT-1 in the context of arboviral infections are lacking, IL-6 is well documented in connection with severe dengue and cardiac issues such as myocarditis [80,81]. This suggests that CT-1 might be involved in similar inflammatory pathways in this arboviral disease.

The role of LDH-B in acute chikungunya still needs to be defined. However, previous studies have noted the presence of LDH isoenzymes during chikungunya infection, suggesting an association with tissue damage or metabolic stress [82,83]. A recent study with Colombian children and adults showed a significant association between CKMB and FABP-3 in fatal cases of chikungunya, which was negative in cases with favorable clinical outcomes [84]. These findings contribute to understanding how chikungunya influences cardiac biomarkers and systemic inflammation. Further studies are needed to clarify the exact role of LDH-B in chikungunya virus infection.

In contrast, when we examined individuals with evidence of acute exposure (anti-IgM) to dengue and Zika, we found no significant association with cardiac and inflammatory biomarkers. This may be due in part to the low rate of acute dengue exposure in our sample and the fact that dengue does not progress to the chronic phase in most cases. Exposure to Zika virus also showed no association with these biomarkers, which could indicate other mechanisms of pathogenicity, such as its neurotropic properties that mainly affect the nervous system [85].

In the Indigenous Fulni-ô population with prior but non-acute exposure to chikungunya, we identified significant associations with elevated levels of big ET-1 and LDH-B. Notably, big ET-1 was also associated with age and chronic kidney disease, but LDH-B was not associated with other comorbidities or age in this study. Big ET-1, a precursor to the potent vasoconstrictor ET-1, is widely recognized for its role in vascular dysfunction and inflammatory processes such as autoimmune arthritis [86].

LDH-B, an isoform predominantly expressed in aerobic tissues such as cardiac muscle, is more specific than total LDH in indicating cardiac stress and injury [87]. Its elevation, alongside big ET-1, may suggest a connection to the chronic inflammatory forms of chikungunya, which are characterized by increased metabolic and cardiac demands. While further research is needed, these findings underscore the potential role of these biomarkers in detecting subclinical cardiac involvement and systemic inflammation in populations exposed to arboviruses [88].

Finally, using the logistic regression model, CT-1 and LDH-B (measured in mean fluorescence intensity) remained significantly associated with chikungunya serological status in the acute phase (IgM) and chronic phase (IgG), respectively. Although CT-1 is a biomarker already known to be associated with myocardial hypertrophy and hypertension, no biomarker was associated with this comorbidity in this study. Accordingly, the role of CT-1 in the acute form of chikungunya as a possible predictor of cardiac damage needs to be investigated in Indigenous populations [89]. Therefore, it is necessary to explore this topic further with more translational studies in order to obtain this information and tools for public health strategies in the near future.

Our study has limitations, mainly due to its cross-sectional design, which precludes the establishment of causal relationships. The sample size was relatively small, particularly with regard to acute arbovirus exposure, which limits our ability to fully study the acute phase of the infections. We were unable to conduct a comprehensive clinical survey to assess the severity of acute illness or to include additional tools to assess subclinical cardiac disease, such as global longitudinal strain. Despite these limitations, it is important to emphasize the pioneering and innovative nature of this study. We analyzed cardiac inflammatory biomarkers using advanced Luminex xMAP technology in a vulnerable Indigenous population undergoing urbanization and exposed to severe health conditions, providing important insights into their cardiovascular health.

## 5. Conclusions

In conclusion, this study showed a high and worrying seroprevalence of dengue, chikungunya, and Zika and provided valuable insights into the association of cardiac biomarkers with serological status for arboviruses (acute and past exposure) in an Indigenous community in the Northeast Region of Brazil, affected by urbanization but still preserving some of their original characteristics. In addition, we also observed essential associations of these cardiac biomarkers and with clinical, anthropometric, sex, and age-related factors. These results underscore the urgent need for future longitudinal studies and broader epidemiologic investigations to deepen our understanding of these associations so that we can develop public health strategies in Indigenous groups.

## Figures and Tables

**Figure 1 viruses-16-01902-f001:**
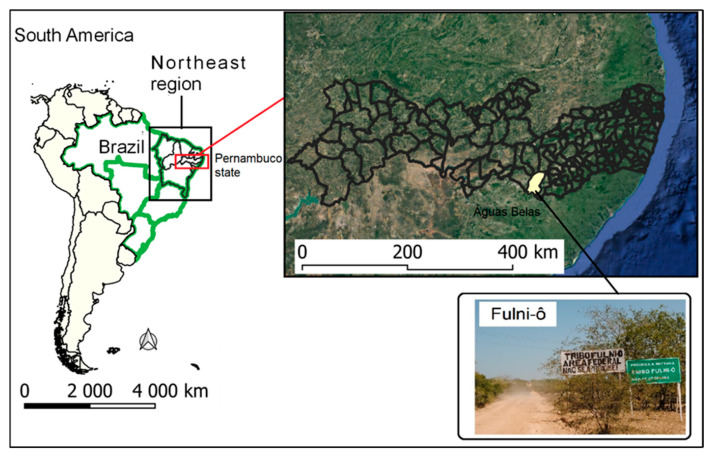
Location of the study area. Fulni-ô ethnic group, Águas Belas, Pernambuco, Brazil.

**Figure 2 viruses-16-01902-f002:**
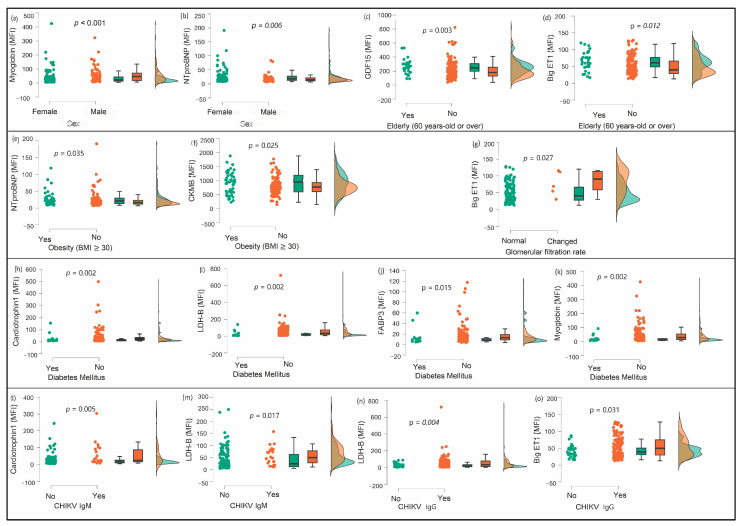
Distribution and comparison of cardiac and inflammatory biomarkers in relation to demographic, anthropometric, and clinical variables with statistical significance. Legend—BMI: body mass index; CKMB: creatine kinase MB; big ET-1: big endothelin 1; FABP-3: fatty-acid-binding protein 3; GDF-15: growth differentiation factor 15; LDH-B: lactate dehydrogenase B; MFI: mean fluorescence intensity; NT-proBNP: N-terminal pro-B-type natriuretic peptide. Comparisons include (**a**,**b**) sex-specific differences in myoglobin and NT-proBNP; (**c**,**d**) age-related changes in GDF-15 and big ET-1; (**e**,**f**) biomarkers associated with obesity (NT-proBNP and CKMB); (**g**) glomerular filtration rate and big ET-1; (**h**–**k**) associations of diabetes mellitus; and (**l**–**o**) CHIKV serology (IgM and IgG) with biomarkers such as cardiotrophin 1, LDH-B, and FABP-3. Statistically significant *p*-values (<0.05) are indicated above the corresponding panels.

**Table 1 viruses-16-01902-t001:** Study population characterization. Fulni-ô, Pernambuco, Brazil (n = 174).

Variable	All *N* = 174	Female *N* = 114	Male *N* = 60	*p*-Value
Age, years (median; IQR)	47 (39–56)	47 (39–57)	47 (38–55)	0.804 ^1^
30 to 39 years	28.2% (49/174)	26.3% (30/114)	31.7% (19/60)	0.475 ^2^
40 to 49 years	27.6% (48/174)	28.9% (33/114)	25.0% (15/60)	
50 to 59 years	27.6% (48/174)	23.7% (27/114)	30.0% (18/60)	
≥60 years (elderly)	18.4% (32/174)	21.0% (24/114)	13.3% (8/60)	
BMI ≥ 30 kg/m^2^ (obesity)	29.3% (51/174)	31.6% (36/114)	25.0% (15/60)	0.365 ^2^
Arterial hypertension	17.8% (31/174)	22.8% (26/114)	8.3% (5/60)	0.018 ^2^
Diabetes mellitus	10.3% (18/174)	12.3% (14/114)	6.7% (4/60)	0.248 ^2^
Chronic kidney disease (eGFR stage ≥ G3)	3.4% (6/174)	4.4% (5/114)	1.7% (1/60)	0.317 ^2^
DENV IgM+	4.6% (8/174)	3.5% (4/114)	6.7% (4/60)	0.333 ^2^
DENV IgG+	92.5% (161/174)	90.3% (103/114)	96.7% (58/60)	0.230 ^2^
CHIKV IgM+	12.6% (22/174)	12/114 (10.5%)	16.7% (10/60)	0.260 ^2^
CHIKV IgG+	78.2% (136/174)	78.0% (89/114)	78.3% (47/60)	0.775 ^2^
ZIKV IgM+	16.1% (28/174)	18.42% (21/114)	11.7% (7/60)	0.212 ^2^
ZIKV IgG+	95.4% (166/174)	96.5% (110/114)	93.3% (56/60)	0.988 ^2^

Legend—^1^: Mann–Whitney U test; ^2^: chi-squared test with continuity correction for categorical variables. BMI: body mass index; CHIKV: chikungunya virus; DENV: dengue virus; eGFR: estimated glomerular filtration rate; IQR: interquartile range; ZIKV: Zika virus. *p*-values in the right column represent the statistical significance of differences between males and females for each variable.

**Table 2 viruses-16-01902-t002:** *p*-values from the Mann–Whitney U test assessing associations between biomarker concentrations (measured in MFI) and sociodemographic, clinical, and serological variables.

	Sociodemographic Variables		Comorbidities				Arboviruses					
Biomarker	Sex Male (n = 114) Female (n = 60)	Age (≥60 Years) Yes (n = 32) No (n = 142)	Obesity (BMI ≥ 30 kg/m^2^) Yes (n = 51) No (n = 123)	Arterial Hypertension Yes (n = 31) No (n = 143)	Diabetes Mellitus Yes (n = 18) No (n = 154)	Chronic Kidney Disease Yes (n = 6) No (n = 128)	DENV IgM+ Yes (n = 8) No (n = 166)	DENV IgG+ Yes (n = 161) No (n = 158)	CHIKV IgM+ Yes (n = 22) No (n = 152)	CHIKV IgG+ Yes (n = 163) No (n = 11)	ZIKV IgM+ Yes (n = 28) No (n = 146)	ZIKV IgG+ Yes (n = 166) No (n = 8)
Cardiotrophin-1 (MFI)	0.587	0.220	0.619	0.208	0.007 *	0.415	0.865	0.193	0.005 *	0.055	0.134	0.592
GDF-15 (MFI)	0.890	0.003 *	0.268	0.631	0.882	0.107	0.651	0.140	0.383	0.136	0.250	0.350
LDH-B (MFI)	0.437	0.781	0.968	0.457	0.002 *	0.868	0.861	0.910	0.017 *	0.004 *	0.910	0.872
FABP-3 (MFI)	0.134	0.277	0.468	0.296	0.015 *	0.448	0.124	0.471	0.433	0.460	0.578	0.588
Myoglobin (MFI)	0.001 *	0.769	0.780	0.562	0.002 *	0.884	0.197	0.053	0.996	0.188	0.800	0.471
NT-proBNP (MFI)	0.006 *	0.306	0.035 *	0.621	0.051	0.889	0.361	0.376	0.496	0.848	0.326	0.821
Troponin I (MFI)	0.555	0.844	0.682	0.276	- ^1^	- ^1^	0.547	0.648	0.826	0.357	0.058	0.893
Big ET-1 (MFI)	0.310	0.012 *	0.272	0.080	0.072	0.027 *	0.331	0.078	0.586	0.031 *	0.478	0.986
CKMB (MFI)	0.525	0.719	0.025 *	0.829	0.348	0.671	0.671	0.493	0.451	0.106	0.426	0.104

Legend—^1^ For variables where variance equals 0, statistical analysis was not performed (-). BMI: body mass index; CHIKV: chikungunya virus; CKMB: creatine kinase MB; DENV: dengue virus; ET-1: endothelin 1; FABP-3: fatty-acid-binding protein 3; LDH-B: lactate dehydrogenase B; GDF-15: growth differentiation factor 15; MFI: mean fluorescence intensity; NT-proBNP: N-terminal pro-B-type natriuretic peptide; ZIKV: Zika virus. * Indicates statistical significance (*p* < 0.05).

**Table 3 viruses-16-01902-t003:** Association between chikungunya serological status (IgM and IgG) and cardiac biomarkers (measured in MFI) using generalized least squares.

**(a) Chikungunya (IgM)**					
	Estimate	Standard error	z	p	VIF
(Intercept)	−0.833	1.443	−0.577	0.564	-
Cardiotrophin 1 (MFI)	0.016	0.006	2.67	0.008 *	1.196
GDF-15 (MFI)	−0.005	0.004	−1.205	0.228	2.337
LDH-B (MFI)	0.009	0.006	1.362	0.173	1.353
FABP-3 (MFI)	−0.055	0.042	−1.298	0.194	1.677
Myoglobin (MFI)	−21.39	0.008	−0.022	0.982	1.541
NT-proBNP (MFI)	−0.007	0.015	−0.458	0.647	1.068
Troponin I (MFI)	−0.091	0.199	−0.456	0.648	1.170
Big ET-1 (MFI)	0.009	0.015	0.601	0.548	2.886
CKMB (MFI)	1.870 × 10^−4^	8.937 × 10^−4^	0.209	0.834	1.252
**(b) Chikungunya (IgG)**					
	Estimate	Standard error	z	p	VIF
(Intercept)	−0.868	1.075	−0.808	0.419	-
Cardiotrophin 1 (MFI)	−0.005	0.005	−1.005	0.315	1.219
GDF-15 (MFI)	5.645 × 10^−4^	0.003	0.194	0.847	2.342
LDH-B (MFI)	0.022	0.009	2.569	0.01 *	1.254
FABP-3 (MFI)	−0.022	0.015	−1.457	0.145	1.671
Myoglobin (MFI)	0.002	0.005	0.365	0.715	1.690
NT-proBNP (MFI)	−60.34	0.008	−0.068	0.946	1.099
Troponin I (MFI)	0.018	0.146	0.122	0.903	1.230
Big ET-1 (MFI)	0.015	0.015	1.008	0.313	3.038
CKMB (MFI)	0.001	7.374 × 10^−4^	1.468	0.142	1.134

Legend—* Indicates statistical significance (*p* < 0.05). CKMB: creatine kinase MB; ET-1: endothelin 1; FABP-3: fatty-acid-binding protein 3; GDF-15: growth differentiation factor 15; LDH-B: lactate dehydrogenase B; MFI: mean fluorescence intensity; NT-proBNP: N-terminal pro-B-type natriuretic peptide; VIF: variance inflation factor. Statistically significant *p*-values (<0.05). Residuals with a normal distribution in both models. The multiple regression analyses for chikungunya (IgM and IgG) used the same sample sizes as indicated in Table 2.

## Data Availability

The datasets generated and/or analyzed during this study are not publicly available due to the protection of participant confidentiality in accordance with Brazilian law. For inquiries regarding the datasets or requests for additional analyses, please contact the corresponding author.

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
