# Peer review of "Cardiac Biomarkers in a Brazilian Indigenous Population Exposed to Arboviruses: A Cross-Sectional Study"

_viruses, 2024, doi:10.3390/v16121902_

Round 1

Reviewer 1 Report

Comments and Suggestions for Authors

The paper looks at biomarkers in an indigenous Brazilian population, and would appear to confirm previous publications regarding biomarkers for atypical/severe disease manifestation of acute CHIKV infections. Bit unclear is whether these associations are more pronounced in this population?

Arboviruses – particularly ZIKV are not only associated with acute disease (line 56 and elsewhere) but are also associated with life-long disabilities (congenital ZIKA syndrome) and protracted rheumatic disease for CHIKV.  Perhaps more importantly given the findings, CHIKV can also present with a complex spectrum of severe disease manifestations and atypical presentations, particularly in the elderly and individuals with comorbidities (reviewed in Nat Rev Rheumatol. 2019 Oct;15(10):597-611) including obesity (Clin Rheumatol. 2024 Sep;43(9):2993-3003) (line 263). 

The most solid associations for CHIKV +ves would appear to be Lactate dehydrogenase B and Cardiotrophin 1 (Tables 2 & 3).  Would be useful to illustrate whether this does nor does not associate with (multiple) comorbidities and/or age.  The simplest explanation is that LDH is associated with severe CHIKV (Emerg Infect Dis. 2016 May;22(5):891–894), so did these patients have disease manifestations and/or comorbidities that might account for elevated LDH?  Would also be helpful to understand how LDH differs from LDH-B.  Cardiotrophin-1 serves as a biomarker of left ventricular hypertrophy and dysfunction in hypertensive patients, with cardiovascular manifestations commonly reported for CHIKV (F1000Res. 2017 May 2;6:390;  Trop. Med. Infect. Dis. 2021, 6(3), 108;  Cardiology 2021;146:324–334) and hypertension a comorbidity well recognised for association with poor outcomes in CHIKV patients (Transactions of The Royal Society of Tropical Medicine and Hygiene 2019, 113, Issue 4, 221–226;  Int J Infect Dis. 2018 Feb:67:107-113).  Thus given these biomarkers are already associated with CHIKV disease (this literture might be included in the manuscript), it is unclear what new information is being provided in this paper – unless perhaps this association is more robust in this population?

The interesting data is shown in Table 2 &3, but this needs to supply n (number of individuals tested and the number of controls tested) for each test and each – what I assume – p value (although this is not stated clearly for Table 2).  All the numbers in Table 2 are in fact p values? The legend says they are biomarker concentrations (line 294) and line 297 refers to Fig. 2 but appears to be a legend for Table 2?  Line 307 is another legend for Fig. 2?  Also unclear for Table 2 are 0.32 over 6 or 0.58 over 8 etc (or is this inappropriate compression of column width for ZIKV?).  

L279.  Please provide time of year (season) and whether low IgM is associated with a dry season and low mosquito numbers. 

Fig. 2 is impossible small font and needs to be made much clearer. What statistical method was used here?

Author Response

Por favor, veja o anexo.

Reviewer 2 Report

Comments and Suggestions for Authors

Cytokines are known to mediate interaction of 3 protection systems of vertebrates (immune, nervous and hormonal). No doubt unspecific innate immunity should be associated with widely spread cardiovascular pathologies. However, currently the direct scientific evidences are limited. Therefore, the multiplex correlation analysis based on highly sensitive imminofluorescent assays with magnetic beads is of interest for researchers and physicians.

General comments.

1. Arboviruses cause not only fever ("febrile illnesses") but severe damages of central nervous system with lethal outcomes.

2. Introduction, lines 56-57.

Epidemiological data are evidently overestimated. Just compare the world's population near 8.025 billion (2023) (30% infection rate with different arboviruses is supposed to be appr. 2,4075 billion) and Dengue virus infection detected in 400 million. 

Lines 111-112. What means "an  indigenous population exposed to arboviruses (DENV, CHIKV, and ZIKV)". How to measure the exposure to arboviruses? How to differentiate between "indigenous population" and newcomes who arrived relatively recently in the endemic regions?  Probably, the total population of endemic regions? 

Methods - ELISA, lines 230-231.

Commonly, TMB is used for ELISA. Spectrofotometer permits to detect optical units in each well of plates. What stands for a "relative index"? It is not widely used. Is it a ratio of optical densities of samples to blank or something else? Explanations are required for the following statement  

"relative index ≥ 1.1 were considered positive. Values ≥ 0.8 and < 1.1 were considered borderline, and values < 0.8 were considered negative". 

Table 1. 

What show P values in the right column? Differences between males and females in each line of the table? 

Why flavivirus-specific IgG were detected in more than 90% of sera studied? It's unbelievably high values. Specific IgG appear in 7-10 days postinfection and later their titers decrease until undetectable in ELISA levels. 

Table 2 

According to the legend the table 2 shows "biomarker concentrations". But measurement units are not described in notes. Probably, some statistical analysis is demonstrated. 

Table 3. 

Association or correlation?

Correlation coefficients or something else are shown in the table 3? 

Disscusion is too long without focus on the main achievements. 

Conclusion contains generalities, abandoning serious correlation analysis between cytokine detection and cardiovascular pathologies according to the title and the goal of the research manuscript.

Specific comments are marked in yellow in the attached file. 

The absence of abbreviation list, introduction of each abbreviation in the text and so on. 

Comments on the Quality of English Language

The English style might be improved. Scientific terms should be used more carefully. The meaning of some statements is not always clear. 

Reviewer 3 Report

Comments and Suggestions for Authors

This is a cross-sectional study of 174 participants in Fulni-ô community (indigenous population) undergoing urbanization in Brazil evaluating cardiac associated cytokines and cardiac markers with clinical and anthropometric factors using Luminex xMAP technology. The authors found that these biomarkers were linked to age, obesity, chronic kidney disease and previous/ recent exposure to Chikungunya.

I really liked that the authors stressed that the research process respected indigenous religious practices and rights and emphasized in the ethical approval.

I look forward to follow-up studies to investigate if certain cut-offs of the mentioned biomarkers are able to prognosticate outcomes on a long term basis and thus would have real-world implications to prompt more intense control of risk factors.

A few comments as follows:

Major comments:

Line 94: “role of arboviruses as potential triggers or contributors to cardiovascular diseases, including subclinical conditions, remains largely unexplored” – there has been some literature on cardiac involvement with arboviruses and “long dengue” with cardiac complications

·         10.1016/j.tcm.2020.11.003

·         10.1371/journal.pntd.0010864

·         10.1159/000514206

·         10.1093/jtm/taae081

Recommend that the authors review the above and a more updated literature review and reference additional studies as needed.

Figure 2: Description should include the different type of graphs plotted

Line 322-337: Given the disparity in arboviral serology – it would be good if the authors put forth some hypothesis or possible reasons for the differences. Were the Fulni-ô community participants for this study older in general, resulting in higher exposure?

Line 337-339 I’m not sure what the authors mean by “it is possible to infer that the epidemiological behavior of these arboviruses in the Fulni-ô indigenous community follows a particular dynamic” aside that it is different. Suggest authors state plainly the intended meaning.

Will recommend rearranging the discussion and placing the most significant part of the study higher in the discussion – i.e. the association with previous / recent exposure to Chikungunya especially since this manuscript title is “Cytokines and cardiac biomarkers in the Brazilian indigenous population exposed to arboviruses”. The association with other demographic variables etc are not the focus of the study.

Line 450: Given the cross-sectional nature of this study, I’m not too sure what the authors mean by “complex dynamics”. As there is only one timepoint, suggest rephrasing.

Minor

·         ET-1 has also been implicated in auto-immune arthritis and one of the long term complications of chikungunya is the development of arthritis – recommend that authors mention this.

Round 2

Reviewer 1 Report

Comments and Suggestions for Authors

Most of the changes are OK but the lack of information on n remains in my view a serious problem for the Tables 2 & 3, n or an n range should be supplied somewhere - perhaps in the legend.  At present it remains unclear whether the statistics are based on e.g. n=2 vs. 3 or 100 vs. 45 (by example), the latter is clearly complelling, the former not. 

Reviewer 2 Report

Comments and Suggestions for Authors

The manuscript was significantly revised but still contains minor inaccuracies marked in the attached file. The main concern is the absence of explanation of discrepancy between high IgG detection rate up to 95% and low IgM levels. Epidemiological data seem to be overestimated. 

Comments on the Quality of English Language

The English language might be corrected especially at the beginning of "Introduction". 
